# Impact of Mutation Profile on Outcomes of Neoadjuvant Therapy in GIST

**DOI:** 10.3390/cancers17040634

**Published:** 2025-02-14

**Authors:** Mahmoud Mohammadi, Evelyne Roets, Roos F. Bleckman, Astrid W. Oosten, Dirk Grunhagen, Ingrid M. E. Desar, Han Bonenkamp, Anna K. L. Reyners, Boudewijn van Etten, Henk Hartgrink, Marta Fiocco, Yvonne Schrage, Neeltje Steeghs, Hans Gelderblom

**Affiliations:** 1Department of Medical Oncology, Leiden University Medical Center, 2300 RC Leiden, The Netherlands; 2Department of Medical Oncology, The Netherlands Cancer Institute, 1066 CX Amsterdam, The Netherlands; e.roets@nki.nl (E.R.);; 3Department of Medical Oncology, University Medical Center Groningen, University of Groningen, 9713 GZ Groningen, The Netherlands; 4Department of Medical Oncology, Erasmus MC Cancer Institute, 3015 CN Rotterdam, The Netherlands; 5Department of Surgical Oncology and Gastrointestinal Surgery, Erasmus MC Cancer Institute, 3015 CN Rotterdam, The Netherlands; 6Department of Medical Oncology, Radboud University Medical Center, 6525 GA Nijmegen, The Netherlands; ingrid.desar@radboudumc.nl; 7Department of Surgery, Radboud University Medical Center, 6525 GA Nijmegen, The Netherlands; 8Department of Surgery, University Medical Center Groningen, University of Groningen, 9713 GZ Groningen, The Netherlands; 9Department of Surgery, Leiden University Medical Center, 2300 RC Leiden, The Netherlands; 10Mathematical Institute, Leiden University, 2300 RA Leiden, The Netherlands; 11Princess Máxima, Center for Pediatric Oncology, 3584 CS Utrecht, The Netherlands; 12Department of Biomedical Data Science, Section Medical Statistics, Leiden University Medical Center, 2333 ZC Leiden, The Netherlands; 13Department of Surgical Oncology, The Netherlands Cancer Institute, 1066 CX Amsterdam, The Netherlands

**Keywords:** neoadjuvant imatinib in GIST, mutational profile in treatment of GIST

## Abstract

Gastrointestinal stromal tumors (GISTs) are rare tumors of the digestive tract, often treated with neoadjuvant imatinib to improve surgical outcomes. However, the response to imatinib varies depending on the tumor’s genetic mutations. This study evaluates how different mutational profiles influence treatment response and surgical outcomes in GIST patients receiving neoadjuvant imatinib. Our findings indicate that patients with KIT exon 11 mutations exhibit a higher rate of partial response and more favorable surgical outcomes compared to those with other mutations. In contrast, patients with non-KIT exon 11 mutations and wild-type KIT/PDGFRA GISTs exhibited less favorable responses to imatinib, suggesting that in these cases, the transition to surgery should occur earlier if there is insufficient therapeutic response. These results highlight the importance of mutational profiling in guiding treatment decisions and optimizing outcomes for GIST patients undergoing neoadjuvant therapy.

## 1. Introduction

Gastrointestinal stromal tumors (GISTs), the most prevalent subtype of sarcoma, is a rare malignancy, accounting for less than 1% of all malignancies arising in the gastrointestinal tract. Incidence rates of GISTs vary across geographic regions [1], although most population-based studies indicate similar epidemiological patterns of GISTs, with an incidence of 15 per million per year [2]. At the time of diagnosis, the median age of patients is in the mid-60s, with a slight male predominance. While GISTs can manifest along the entire gastrointestinal tract, the majority are found in the stomach (56%) and the small intestine (32%), whereas localization in the colon (6%) and rectum (1%), esophagus (<1%), and other less common sites (4%) is very uncommon [3]. The crucial discovery of KIT protein overexpression, a receptor tyrosine kinase also known as CD-117, revolutionized understandings and diagnosis of GISTs as an entity, over two decades ago [4]. Simultaneously, the identification of gain-of-function mutations leading to uncontrolled KIT activation represented a breakthrough in clinical practice. These mutations, occurring in the KIT or platelet-derived growth factor receptor alpha (PDGFRA) genes, drive aberrant signaling cascades, resulting in sustained growth, proliferation, and inhibition of apoptosis, ultimately leading to the development of GIST [4,5]. Approximately 80% of GISTs arise from oncogenic KIT mutations, whereas PDGFRA mutations are responsible for 10–15% of cases. The remaining 5–10% of GISTs, formerly referred to as wild-type GISTs, exhibit other stigmata such as succinate dehydrogenase (SDH) deficiency, neurofibromatosis type 1 (NF1) mutations, occasional neurotrophic receptor tyrosine kinase (NTRK) and BRAF mutations [6,7,8,9].

Surgical resection remains the primary treatment approach for patients with localized GISTs [10]. In patients with large or locally advanced tumors, neoadjuvant therapy with imatinib is common practice. Downsizing the tumor and converting surgical irresectable or locally advanced GISTs to resectable GISTs, is the primary goal of neoadjuvant therapy [11,12,13]. In addition, neoadjuvant imatinib treatment can help to avoid perioperative spill and to achieve negative surgical margins (R0 resection) [14], both of which are associated with better long-term outcomes [11]. Furthermore, in cases where GISTs are located in specific anatomical sites (e.g., esophagus, pancreas, and rectum), an important goal of neoadjuvant treatment is to preserve organ function by enabling a less invasive surgical procedure and preventing unnecessary reconstructions [15].

While the therapeutic efficacy of imatinib is well established, there is a limited understanding of how specific mutational profiles influence treatment response and surgical outcomes. The heterogeneity of mutations in GIST may introduce variability in response to therapy, timing of optimal surgical intervention and the achievement of negative surgical margins. These knowledge gaps hinder the development of individualized treatment strategies and may lead to suboptimal patient outcomes. We hypothesize that mutational profiles significantly influence the clinical and surgical outcomes of patients with GISTs undergoing neoadjuvant imatinib therapy. The objective of this study is to evaluate the clinicopathological features, treatment responses, and surgical outcomes in relation to the mutational profiles of GISTs.

## 2. Methods

### 2.1. Patients and Study Design

In a retrospective study, patients with histologically proven GISTs, diagnosed between January 2009 and May 2021, were included. Data were obtained from the Dutch GIST Registry (DGR), a prospective real-life database containing clinical data of all GIST patients treated in five GIST specialized centers in the Netherlands: Netherlands Cancer Institute, Leiden University Medical Center, Erasmus Medical Center, Radboud University Medical Center, and University Medical Center Groningen. The inclusion criteria were age 18 years or older, having a reported GIST-specific mutational analysis, and treatment with neoadjuvant imatinib. Neoadjuvant imatinib was initiated in patients, when tumor size, location of the GIST or patient condition suggested that preoperative therapy could facilitate a less invasive or lower-risk surgical approach, which is defined as surgical techniques that minimize surgical morbidity and reduce perioperative risks compared to traditional open surgeries. These approaches aim to achieve complete tumor resection while decreasing the extent of resection; therefore, preserving organ function. In addition, neoadjuvant imatinib was advised in patients for primary localized GISTs, initially assessed as inoperable. All patients undergoing neoadjuvant therapy were evaluated in advance by a multidisciplinary team, comprising sarcoma specialists from oncology, surgery, radiology, and pathology. The local Medical Ethics Review Committee of LUMC confirmed that the Medical Research Involving Human Subjects Act did not apply for this study (reference number nWMODIV2_2022003). Furthermore, this multicenter study was approved by the local review boards of centers participating in the national database of patients with GISTs in the Netherlands—the Dutch GIST Registry (DGR) (reference number IRBd20-212).

### 2.2. Variables of Interest

Demographic data, tumor characteristics, radiological findings, and surgical outcomes were included. GIST location was categorized as the esophagus, gastric, duodenum, small bowel, colon, rectum, or mesenterium. Tumor size before and after treatment was measured as the largest transverse diameter in centimeters, based on radiologic assessment. The used radiology modalities were mainly computed tomography (CT) and/or positron emission tomography-computed tomography (PET-CT). In specific cases (e.g., rectal GIST), magnetic resonance imaging (MRI) was used. The stage at diagnosis was categorized as a localized, locally advanced, or metastasized GIST. Response evaluation (i.e., stable disease, partial response, complete response, or progressive disease) according to RECIST 1.1 [16] after initiating neoadjuvant imatinib was included. Locally advanced GISTs were defined as non-metastatic GISTs with extension of the tumor to adjacent structures or when neoadjuvant imatinib was indicated. Mutational status was categorized as *KIT*-exon 9, *KIT*-exon 11, *KIT*-exon 13, *KIT*-exon 17, *PDGFRA*-exon 12, *PDGFRA*-exon 14, *PDGFRA*-exon 18 non-D842V, *PDGFRA*-exon 18 D842V, SDH and NTRK fusion. Patients without a mutation of KIT and PDGFRA, and without SDH mutational analysis were labeled as wild-type KIT/PDGFRA. Patients with wild-type KIT/PDGFRA/SDH were patients without a KIT, PDGFRA, and SDH mutation on mutational testing. Surgical margins were classified as R0, R1, and R2 according to established criteria [17]. The variables of tumor localization, mutational status, and resection margin were recorded as categorical and tumor size as continuous. In addition, response evaluation (i.e., stable disease, partial response, complete response, or progressive disease) after initiating neoadjuvant imatinib was included.

### 2.3. Outcomes

The primary outcome was the response rate of neoadjuvant treatment. The secondary outcomes were negative resection margin (R0) versus positive resection margin (R1 or R2) and the time on neoadjuvant therapy (which reflects the interval from the start of imatinib to the time of surgery). In addition to the primary and secondary outcomes, we assessed the concordance between tumor size reported on preoperative computed tomography (CT) scans and pathological resection specimens, given its potential clinical implications.

In our study, we analyzed the CT scans to facilitate a precise determination of tumor size. The primary and secondary outcomes were compared between patients having a *KIT* exon 11 and patients with a non-*KIT* exon 11 mutation. The response evaluation was assessed according to the Response Evaluation Criteria in Solid Tumors (RECIST) version 1.1 [16] by local investigators. Objective response rate was defined as a partial or complete response. Response evaluation was performed by a CT scan every 3 months, in accordance with (inter)national guidelines [10]. If the patient had symptoms or complaints that might be caused by progression of the GIST, the CT scan was performed earlier.

### 2.4. Statistical Analysis

The Chi-square test was used to compare the response rates and results of resection margin between patients with *KIT*-exon 11 mutation and non-*KIT* exon 11 mutation. Time on neoadjuvant therapy was calculated from the start date to the date of surgical resection. Time on neoadjuvant therapy was compared between groups using t-test. Concordance of tumor size reported on a preoperative CT scan with the tumor size reported on the pathological resection specimen was evaluated and visualized using the Bland–Altman plot. Statistical analysis was performed with IBM SPSS Statistics 28. A *p* value of < 0.05 was labeled as significant.

## 3. Results

### 3.1. Demographic and Clinicopathological Characteristics

A total of 326 GIST patients who were treated with neoadjuvant imatinib were included in this study. Out of these, surgical resection of GISTs was performed in 264 patients, accounting for 80.9% of the total cohort (Figure 1). In the remaining 62 patients (19.1%), surgical treatment was not undertaken due to a variety of reasons, including ongoing neoadjuvant therapy, interference from other concurrent diseases, progression of the GIST itself, the patient’s personal decision, or an assessment that surgery was not feasible based on clinical judgment. The majority of patients, specifically 60.4%, had a gastric GIST. At the time of initial diagnosis, 99.4% of patients presented with either localized or locally advanced disease stages (Table 1). Mutational analysis performed on these patients identified a KIT-exon 11 mutation in 247 (75.8%) individuals, while the remaining patients either had non-KIT-exon 11 mutations or a wild-type GIST. Patients with KIT-exon 11 mutations typically started with a dosage of 400 mg of imatinib, while 72% of those with a KIT-exon 9 mutation received an initial dose of 800 mg. For patients with a PDGFRA-D842V mutation (*n* = 10), who had also started on neoadjuvant imatinib, mutational status was not available at the time imatinib treatment was initiated.

### 3.2. Outcomes

#### 3.2.1. Primary Outcome

Several important prognostic factors, such as tumor size and the anatomical location of the GIST, showed similar distributions across both KIT-exon 11 and non-KIT-exon 11 mutated GIST groups (Table 2). After treatment with neoadjuvant imatinib, stable disease was observed in 71 patients (35.5%) of the KIT-exon 11 group, and in 34 patients (51.5%) of the non-KIT-exon 11 group. Partial response to neoadjuvant imatinib was more frequently noted in patients with KIT-exon 11 mutations compared to those with non-KIT exon 11 mutations, with rates of 60.5% versus 33.3%, respectively (*p* < 0.01). Progressive disease was observed more frequently in the non-KIT-exon 11 group than in the KIT-exon 11 group, with rates of 10.6% and 3.5%, respectively (*p* < 0.01). Specifically, seven patients with a KIT-exon 11 mutation and ten with a non-KIT exon 11 mutation showed disease progression in response to neoadjuvant imatinib, prior to undergoing surgery. Among the seven patients with a KIT-exon 11 mutation who progressed, five had non-gastric GISTs (located in the small bowel, rectum, or duodenum) with a median tumor size of 158 mm (range 53.0–250.0 mm) prior to the initiation of imatinib therapy. In the non-KIT exon 11 mutation group, five out of ten patients with progression had non-gastric GISTs with a median tumor size of 116 mm (range 55.0–200.0 mm) at baseline.

#### 3.2.2. Secondary Outcomes

The median duration of neoadjuvant therapy differed significantly between the two groups, with KIT-exon 11 mutated GIST patients undergoing an average of 8.8 months (range 0.2–31.3 months) of treatment compared to 5.3 months (range 0.5–21.0 months) for patients with non-KIT-exon 11 mutated GIST (*p* < 0.001). Negative resection margins (R0) were achieved in 236 patients, which corresponds to 89.8% of those who underwent resection (Table 3). Positive resection margins (R1 or R2) were identified in 24 patients, while in four cases, margin status was not documented. Positive margins were significantly more common in patients with non-KIT-exon 11 mutations (14 out of 66, or 21.2%) compared to those with KIT-exon 11 mutations (10 out of 198, or 5.5%) (*p* < 0.01).

Among the 14 patients with a non-KIT-exon 11 mutation and positive margins, seven (50%) were identified as KIT/PDGFRA wild-type. Within this group, mutations included PDGFRA-D842V (*n* = 1, 7%), PDGFRA non-D842V (*n* = 2, 14%), KIT-exon 9 (*n* = 2, 14%), KIT-exon 17 (*n* = 1, 7%), and PDGFRA-exon 14 (*n* = 1, 7%). Six of these fourteen patients showed GIST progression despite treatment with neoadjuvant imatinib. The average tumor size in this subgroup was 130 mm, and most of the tumors were located in the small bowel (66.7%). Four of these cases were identified as KIT/PDGFRA/SDH wild-type, one patient (16.7%) had a KIT-exon 9 mutation, and another patient (16.7%) had a PDGFRA-exon 14 mutation.

In the subgroup of patients with a KIT-exon 11 mutation and positive resection margins (*n* = 10), four patients (40.0%) had achieved a partial response, while six patients (60.0%) had stable disease. The median tumor size in this subgroup was 142 mm (range 19.0–300.0 mm), with the tumor located in the stomach for five patients (45.5%), in the small bowel for four patients (36.4%), and in the rectum for two patients (18.1%).

Finally, when comparing tumor size as assessed on preoperative CT scans with the tumor size measured on resected specimens, a notable difference of at least 25% was observed in 75 patients (28.2%) (Figure 2). This analysis was not defined as a primary or secondary outcome but was included as a variable of interest due to its potential clinical relevance.

## 4. Discussion

The treatment of GISTs has revolutionized since the introduction of imatinib, and has led to the transformation of management of GISTs in daily practice, including the implementation of neoadjuvant treatment with imatinib. This study aimed to elucidate the impact of different mutations on the outcomes of neoadjuvant imatinib therapy in GIST patients.

Our results indicated variations in treatment response based on the specific mutational profile. Notably, patients with *KIT* exon 11 mutations showed a higher rate of partial response to neoadjuvant imatinib compared to those with non-*KIT* exon 11 mutations. This finding aligns with previous clinical studies [18,19,20].

Focusing on underlying molecular mechanisms, in contrast to *KIT* exon 11 mutations, wild-type KIT/PDGFRA GISTs rely less on KIT signaling for tumorigenesis, involving alternative oncogenic pathways that limit imatinib’s therapeutic effect [21]. Furthermore, specific PDGFRA mutations, such as D842V in exon 18, induce conformational changes that directly prevent imatinib binding, rendering these tumors inherently resistant [22]. However, *KIT* exon 11 mutations are localized in the juxtamembrane domain of the KIT receptor tyrosine kinase, a critical region that regulates receptor activation. Under normal conditions, the juxtamembrane domain maintains the receptor in an inactive conformation, preventing ligand-independent activation. Mutations in exon 11 disrupt this autoinhibitory function, leading to constitutive activation of the receptor, which eventually drives tumorigenesis by promoting proliferation, survival, and growth in GISTs [4,23]. Imatinib targets the ATP-binding site of the KIT receptor. For effective binding, the receptor must adopt an inactive conformation. KIT exon 11 mutations, despite their constitutive activity, retain the receptor in a conformation amenable to imatinib binding. This results in the efficient blockade of ATP binding, inhibiting tyrosine kinase activity and downstream signaling, ultimately halting tumor growth and inducing apoptosis [24,25]. In contrast, other KIT mutations, such as those in exon 9, occur in the extracellular domain of the receptor. These mutations promote ligand-independent dimerization, stabilizing the receptor in an active conformation. This structural alteration reduces the binding affinity of imatinib to the ATP pocket, thereby decreasing its efficacy [26]; therefore, a daily dose of 800 mg imatinib (standard dose is 400 mg) is recommended for *KIT* exon 9 mutation [19]. In our study, 18 patients (5.5% of the total cohort) were identified as harboring KIT exon 9 mutations. Of these, 13 patients were started on the recommended dose of 800 mg imatinib, while 5 patients received an initial dose of 400 mg. Unfortunately, it was not possible to determine whether the dose was later escalated to 800 mg in these 5 patients. Given the small size of this subgroup, we believe their impact on the overall results to be minimal.

While imatinib-sensitive mutations facilitate the therapeutic action of imatinib, resistance can arise through secondary mutations or the activation of alternative mechanisms that bypass KIT inhibition. Secondary mutations that arise in the kinase domain of the KIT and PDGFRA receptors impair imatinib binding. For example, mutations in KIT exon 13, 14, and 17, particularly the T670I mutation in the activation loop, have been shown to reduce imatinib binding affinity by altering the conformation of the kinase domain, thereby diminishing the drug’s efficacy [27]. Beyond secondary mutations, GISTs can develop resistance through the activation of downstream signaling pathways that bypass KIT inhibition, such as the MAPK/ERK pathway [28]. Another significant resistance mechanism involves the overexpression of efflux transporters. These transporters can actively expel imatinib from tumor cells, reducing its intracellular concentration and thus limiting its therapeutic efficacy [29]. In addition to these mechanisms, tumor heterogeneity is a critical factor to consider when imatinib efficacy diminishes. Within a single tumor, distinct cellular clones may harbor diverse secondary mutations or engage alternative resistance mechanisms [30,31], creating therapeutic challenge that limits the effectiveness of imatinib treatment. This could explain incomplete responses or progressive disease in some patients.

The primary objectives of neoadjuvant imatinib therapy (in GISTs) are to achieve an R0 resection margin and facilitate a less invasive surgical procedure. Our study demonstrated that patients treated with neoadjuvant imatinib achieved a high rate of R0 resection, consistent with previously published data [14,32,33]. We demonstrated that 89.8% of patients who received neoadjuvant imatinib achieved an R0 resection. A previous study [34], involving patients with gastric GISTs, with a tumor size of 10 cm or larger, showed R0 resection in 91% of patients treated with neoadjuvant imatinib. A comparable study involving a similar patient population with gastric GISTs demonstrated an R0 resection rate of 94% after treatment with neoadjuvant imatinib [32]. It is important to note that the aforementioned studies included only patients with gastric GISTs, whereas in our study, 38.2% of patients had non-gastric GISTs. Gastric GISTs are generally associated with a more favorable prognosis compared to non-gastric GISTs [3,35]. In our study, the risk of having an R1/R2 resection was higher in patients with non-gastric (mainly small bowel) GISTs compared to those with gastric GISTs.

Focusing on long-term outcomes, an earlier study showed that R0 resection was associated with improved disease-free survival and overall survival in a cohort of 39 patients with resected rectal GISTs [36]. Studies with larger patient populations and multiple GIST locations have also demonstrated that RO resection and preoperative imatinib treatment led to improved disease-free survival and overall survival [37,38]. In contrast, research conducted on patients receiving adjuvant imatinib revealed that R1 resection did not result in poorer survival outcomes [27]. The correlation between the R0 resection margin and improved long-term outcomes might be under debate due to these contrasting results. However, achieving an R0 resection margin remains a critical goal of neoadjuvant therapy in GISTs. Our study shows the successful accomplishment of this goal in most patients (92% and 79% in *KIT*-exon 11 and non-*KIT* exon 11 mutations, respectively). When evaluating potential risk factors associated with failure to achieve neoadjuvant therapy goals, our data suggest a potential correlation with non-KIT exon 11 mutations, especially wild-type KIT/PDGFRA GIST, and GIST localization in the small intestine. These findings imply the need to consider an early transition from neoadjuvant imatinib to surgical resection in patients with wild-type KIT/PDGFRA GIST located in the small intestine when there is an insufficient response to imatinib.

In addition to achieving an R0 resection, tumor downsizing and its facilitation a less invasive or less extensive organ-preserving surgical approach, is an important goal of neoadjuvant imatinib in GISTs. This goal was achieved in a previous study involving 40 patients with locally advanced GISTs (localizations in the stomach, small intestine, and rectum) treated with neoadjuvant imatinib, where organ-preserving surgery could be performed in 70.4% of patients who initially were scheduled for a full-organ resection (e.g., total gastrectomy or low anterior resection) [39]. The success rate of organ-preserving surgeries was even higher in a study focusing solely on gastric GISTs, with 94.6% of patients undergoing partial gastrectomy instead of total gastrectomy following neoadjuvant imatinib treatment [34]. In addition to promoting organ-saving approaches through objective tumor downsizing, neoadjuvant imatinib should be considered particularly in patients where the initial surgical approach is associated with a high risk of complications such as tumor rupture or significant intraoperative hemorrhage. This is due to the fact that it enhances the integrity of the tumor capsule and reduces the risk of tumor rupture or intraperitoneal bleeding [14,40]. Tumor rupture represents a significant risk factor in GISTs. Multiple studies [21,41,42] indicate high recurrence rates and poorer survival outcomes after the occurrence of tumor rupture (either spontaneous or intraoperative). Moreover, patients who experience tumor rupture are at high risk of developing peritoneal metastases. In GISTs, it has been demonstrated that individuals with peritoneal metastases have a poorer prognosis compared to those with hepatic metastases [43]. These factors, highlight the importance of neoadjuvant imatinib in minimizing the likelihood of tumor rupture. It is not known whether mutation profiles influence the risk of tumor rupture when treated with neoadjuvant imatinib. However, neoadjuvant imatinib has been shown to reduce tumor size and the extent of surgery, potentially preventing surgical morbidity. Given the observation in this study, that KIT exon 11 mutations often induce a significant response to imatinib, this highlights the importance of treating patients with unresectable or locally advanced GIST-harboring KIT exon 11 mutations with neoadjuvant imatinib. For patients with non-KIT exon 11 or wild-type KIT/PDGFRA mutations associated with lower response rate in our study, an earlier transition to surgical resection might be warranted, if neoadjuvant imatinib does not appear to induce an evident response. These considerations emphasize the necessity of integrating molecular profiling into the decision-making process for borderline resectable GISTs.

We observed that patients with non-*KIT*-exon 11 mutations had a significantly shorter duration of neoadjuvant treatment compared to those with *KIT*-exon 11 mutations. This discrepancy can be explained by the finding that non-*KIT*-exon 11 mutations were less responsive to imatinib in our study, consequently leading to a more rapid switch from neoadjuvant treatment to surgical resection. Following this observation, it should be noted that when early response assessment (6–8 weeks after initiation of neoadjuvant imatinib) is warranted, a PET-CT may offer valuable information in the context of non-*KIT* exon 11 mutations, as elucidated by a prior investigation [44].

CT is the most widely utilized method for assessing response in GISTs, with the most extensive experience reported in the literature [45]. In clinical practice, preoperative CT scans are used to assess the extent of the tumor and its relationship with nearby structures, and help in surgical planning. After the surgery, the tumor size is measured by the pathologist using macroscopic examination of the resected tissue. This measurement provides a direct and precise assessment of the tumor’s dimensions. In our study, a significant discrepancy between tumor size assessed by the preoperative CT scan and tumor size measured on the resection specimen measured by the pathologist, was observed in 28.2% of patients. While the interval between preoperative assessment to surgery was similar between patients, factors such as edema and inflammation around the tumor, the irregular shape of GISTs and cystic transformation may affect the accuracy of tumor size measurements. The tumor size measured on CT is not always indicative of the true dimensions.

Several limitations should be acknowledged in this study. The retrospective nature of the data collection could introduce biases and confounding variables, such as selection bias or incomplete data collection inherent to this design. The difference between the sample size of the KIT-exon 11 and non-KIT-exon 11 groups may influence the interpretation of results, particularly in subgroup analyses. Furthermore, the lack of data on recurrence and survival in our study, due to relatively short follow-up, limits the possibility of drawing conclusions on the long-term effects of different mutational profiles. This absence of survival data restricts insights into how mutational status might influence outcomes beyond the neoadjuvant phase. Additionally, adverse events were not included, which prevents assessment of the potential toxicity profile of neoadjuvant imatinib in this cohort. Nevertheless, this study is unique to our knowledge with a substantial sample size and provides detailed information on the mutational status of GIST patients treated with neoadjuvant imatinib in everyday clinical settings.

Our findings might have significant implications for clinical practice. Based on the observed treatment responses, it is clear that mutation testing should guide the decision to transition from neoadjuvant therapy to surgery. Our study found that patients with non-KIT exon 11 mutations and wild-type KIT/PDGFRA GISTs exhibited less favorable responses to imatinib, suggesting that in these cases, the transition to surgery should occur earlier if there is insufficient therapeutic response. Specifically, for these patients, early assessment of response to neoadjuvant therapy (by using PET-CT imaging) after 6–8 weeks of treatment may provide valuable information to guide clinical decision-making. If a limited response or no response is observed, an earlier switch to surgical resection might be warranted, particularly to avoid tumor progression.

## 5. Conclusions

In conclusion, our study provides valuable insights into the impact of different mutations on neoadjuvant imatinib therapy outcomes in GIST patients. The observed differences in treatment response, resection margins, and treatment duration highlight the impact of mutation profiles on outcomes of neoadjuvant imatinib.

## Figures and Tables

**Figure 1 cancers-17-00634-f001:**
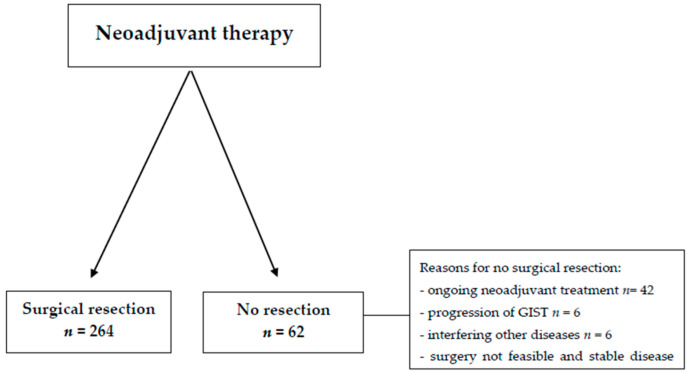
Flowchart.

**Figure 2 cancers-17-00634-f002:**
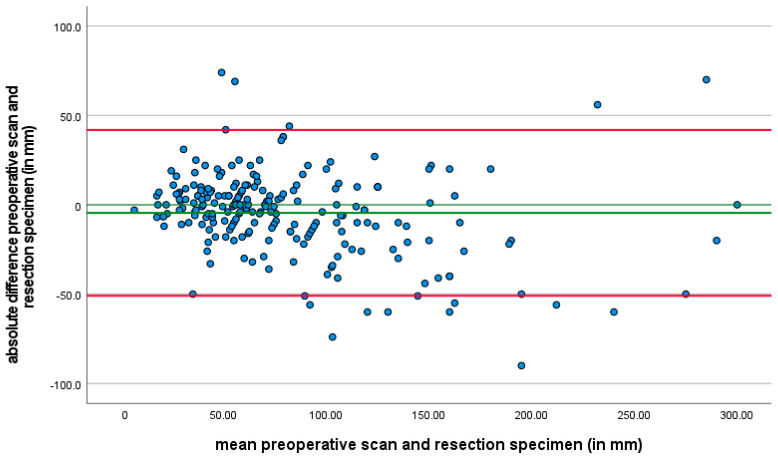
A Bland–Altman plot is provided to illustrate the agreement between tumor size assessed by preoperative CT scans and tumor size measured on resection specimens. In the plot, the x-axis represents the average of the two measurements (CT scan and resection specimen size), while the y-axis shows the difference between the two measurements for each patient. This Bland–Altman plot highlights the extent of variability between the two methods, and it is evident that there is a considerable range of differences in tumor size assessment across patients. The discrepancies between the tumor size measurements were observed in 28.2% of patients.

**Table 1 cancers-17-00634-t001:** Baseline characteristics.

Characteristics	All Patients No. (%)	Patiens with Resection No. (%)	Patients Without Resection No. (%)
Numbers of patients	326 (100)	264 (80.9)	62 (19.1)
Gender			
Male	180 (55.2)	140 (53.0)	40 (64.5)
Female	146 (44.8)	124 (47.0)	22 (35.5)
Age at diagnosis, mean (SD)	63.1 (12.9)	61.8 (12.6)	68.7 (12.7)
Stage at diagnosis			
Localized	50 (15.3)	40 (15.1)	10 (16.1)
Locally advanced	273 (83.7)	221 (83.7)	52 (83.9)
Multifocal GIST	3 (0.9)	3 (1.1)	0
Tumor size at baseline in mm, median (range)	100.0 (11.0–330.0)	100.0 (19.0–300.0)	90.4 (11.0–330.0)
Primary location			
Gastric	197 (60.4)	166 (62.8)	31 (50.0)
Small bowel	32 (9.8)	27 (10.2)	5 (8.1)
Duodenum	35 (10.7)	30 (11.4)	5 (8.1)
Rectum	50 (15.3)	36 (13.6)	14 (22.6)
Esophagus	7 (2.1)	2 (0.8)	5 (8.1)
Mesenterium	2 (0.6)	2 (0.8)	1 (1.6)
Pelvic cavity	2 (0.6)	0 (0.0)	1 (1.6)
Colon	1 (0.3)	1 (0.4)	0
Mutation			
KIT exon 11	247 (75.8)	197 (74.6)	50 (80.6)
KIT exon 9	18 (5.5)	13 (5.0)	5 (8.1)
KIT exon 13	5 (1.5)	5 (1.9)	0
KIT exon 17	1 (0.3)	1 (0.4)	0
PDGFRA exon 12	2 (0.6)	1 (0.4)	1 (1.6)
PDGFRA exon 14	3 (0.9)	3 (1.1)	0
PDGFRA exon 18 non-D842V	17 (5.2)	14 (5.3)	3 (4.8)
PDGFRA exon 18 D842V	10 (3.0)	10 (3.8)	0
SDH mutation	1 (0.3)	1 (0.4)	0
NTRK fusion	2 (0.6)	2 (0.7)	0
WT KIT/PDGFRA/SDH	8 (2.4)	7 (2.6)	1 (1.6)
WT KIT/PDGFRA	12 (3.7)	10 (3.8)	2 (3.2)
Starting imatinib dose			
400 mg	313	253	60
800 mg *	13	11	2

* 13 of 18 patients with a *KIT* exon 9 mutation were treated with imatinib 800 mg. PDGFRA: platelet-derived growth factor receptor alpha, SDH: Succinate dehydrogenase, NTRK: Neurotrophic tyrosine receptor kinase, WT: wild-type.

**Table 2 cancers-17-00634-t002:** Characteristics of patients with a surgical resection.

Characteristics	KIT Exon 11 No. (%)	Non-KIT Exon 11 No. (%)
Number of patients	198 (75.0)	66 (25.0)
Gender		
Male	135 (54.4)	45 (57.7)
Female	113 (45.6)	33 (42.3)
Age at diagnosis, mean (SD)	63.6 (12.3)	61.5 (14.6)
Stage at diagnosis		
Localized	29 (14.6)	11 (16.7)
Locally advanced	169 (84.5)	52 (78.8)
Multifocal GIST	0 (0)	3 (4.5)
Tumor size at baseline in mm, median (range)	100 (11.0–300.0)	107 (25.0–330.0)
Primary location		
Gastric	120 (60.4)	46 (70.0)
Small bowel	21 (10.4)	6 (9.1)
Duodenum	21 (8.8)	9 (13.6)
Rectum	32 (16.4)	4 (6.0)
Esophagus	2 (2.8)	0 (0.0)
Mesenterium	2 (0.8)	0 (0.0)
Pelvic cavity	0 (0.0)	0 (0.0)
Colon	0 (0.0)	1 (1.3)
Time on neoadjuvant therapy, median (range)	8.8 (0.2–31.3)	5.3 (0.5–21.0)
Resection margin *		
Negative (R0)	184	52
Positive (R1/R2)	10	14

* The resection margin was not reported in four patients.

**Table 3 cancers-17-00634-t003:** Characteristics of patients with negative or positive surgical margin *.

Characteristics	Negative Margin (R0)	Positive Margin (R1)
Number of patients	236 (90.1)	26 (10.9)
Gender		
Male	121 (51.3)	17 (65.4)
Female	115 (48.7)	9 (34.6)
Age at diagnosis, mean (SD)	62.1 (12.9)	58.9 (10.0)
Stage at diagnosis		
Localized	38 (16.1)	1 (3.8)
Locally advanced	195 (82.6)	25 (96.2)
Multifocal GIST	3 (1.3)	0 (0)
Tumor size at baseline in mm, median (range)	112.0 (19.0–250.0)	200.0 (39.0–230.0)
Primary location		
Gastric	153 (64.8)	12 (46.2)
Small bowel	23 (9.7)	4 (15.4)
Duodenum	23 (9.7)	7 (26.9)
Rectum	32 (13.6)	3 (11.5)
Esophagus	2 (0.8)	0 (0)
Mesenterium	2 (0.8)	0 (0)
Colon	1 (0.4)	0 (0)
Response evaluation prior to surgery		
Stable disease	93 (39.4)	10 (38.5)
Partial response	132 (55.9)	10 (38.5)
Complete response	1 (0.4)	0 (0)
Progressive disease	10 (4.2)	6 (23.0)
Mutation		
*KIT* exon 11	184 (78.0)	11 (42.3)
*KIT* exon 9	10 (4.2)	3 (11.5)
*KIT* exon 13	5 (2.1)	0 (0)
*KIT* exon 17	0 (0)	1 (3.8)
PDGFRA exon 12	1 (0.4)	0 (0)
PDGFRA exon 14	2 (0.8)	1 (3.8)
PDGFRA exon 18 non-D842V	12 (5.1)	2 (7.7)
PDGFRA exon 18 D842V	9 (3.8)	1 (3.8)
SDH mutation	1 (0.4)	0 (0)
NTRK fusion	2 (0.8)	0 (0)
WT *KIT*/PDGFRA/SDH	3 (1.3)	4 (15.4)
WT *KIT*/PDGFRA	7 (3.0)	3 (11.5)

* The resection margin was not reported in four patients. PDGFRA: platelet derived growth factor receptor alpha, SDH: Succinate dehydrogenase, NTRK: Neurotrophic tyrosine receptor kinase, WT; wild-type.

## Data Availability

The data presented in this study are available on request from the corresponding author.

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
