# Peer review of "Impact of Mutation Profile on Outcomes of Neoadjuvant Therapy in GIST"

_cancers, 2025, doi:10.3390/cancers17040634_

Round 1
Reviewer 1 Report
Comments and Suggestions for Authors
Major Comments
1. The introduction does not provide the rationale or the importance of studying mutational profiles to overcome the challenges in GIST treatment. Add an adequate explanation of this aspect and connect it to the background. Also, while the paper provides descriptive outcomes based on mutational profiles in GIST patients treated with neoadjuvant imatinib, it does not articulate a clear hypothesis. The objective is presented as "evaluating response and surgical outcomes," but the lack of a central hypothesis weakens the scientific narrative.
2. What do you mean by "less invasive or lower-risk surgical approaches" (what are your criteria for classification? Elaborate more regarding these concepts.
3. In lines 186-187, when presenting a comparison between patients with KIT-exon 11 mutations and with non-KIT exon 11 mutations in terms of disease progression, the authors did not address the huge sample size difference between the two categories, this skews the interpretation of the results.
4. The primary outcome (response rate) and secondary outcomes (R0 margin and therapy duration) are not well-defined. For example, how was a "partial response" determined using RECIST 1.1? Was it independently reviewed or assessed by a single investigator?
5. In lines 201-202, the authors write “Negative resection margins (R0) were achieved in 236 patients, which corresponds to 89.8% of those who underwent resection (Table 3)”. Yet they do not break this percentage between KIT and non-KIT mutation groups.
6. Discussing why the response rates differed significantly across mutational profiles could enrich the “Primary outcome” section.
7. There is somewhat overemphasis on discrepancies between preoperative imaging and resection specimens without signifying its clinical implications.
8. For KIT-exon 11 mutations, data is given in total but do not differentiate tumor location and size. Do gastric KIT-exon 11 GISTs respond better compared to rectal or small bowel?
9. The paper identifies differences in treatment responses based on mutational profiles but does not sufficiently discuss the underlying molecular mechanisms that might explain these variations:
a. Why do KIT exon 11 mutations exhibit better responses and longer durations of neoadjuvant therapy?
b. How do non-KIT exon 11 mutations contribute to resistance to imatinib?
c. The potential role of secondary mutations or resistance pathways is not explored.
10. The paper does not address potential tumor heterogeneity within the same patient, which could influence treatment response and surgical outcomes. For example, the presence of mixed mutational subclones or resistance mutations could explain incomplete responses or progressive disease in some cases.
11. The paper reports differences in initial imatinib doses for certain mutations (e.g., KIT exon 9 patients started on 800 mg), but it does not analyze how dose escalation or reduction might have influenced response rates, duration of therapy, or surgical outcomes.
12. The authors assess the concordance between tumor size reported on preoperative CT scans and pathological resection specimens, and categorize it as a secondary outcome. However, the connection between this analysis and the impact of mutational profiles is not explained anywhere in the manuscript. How is this a relevant outcome to the mutational profile?
13. The study provides valuable insights into mutational profiles and treatment responses but does not sufficiently translate these findings into actionable clinical recommendations. For example:
- Should mutation testing guide decisions on when to transition from neoadjuvant therapy to surgery?
- How should clinicians manage patients with limited or no response to imatinib?
Minor Comments
o The abstract does not explicitly mention the study design (retrospective cohort). Adding this information would provide readers with context upfront.
o “Gastric GISTs are generally associated with a more favorable prognosis compared to non-gastric GISTs.” is not directly supported by any specific citation in the discussion section.
o In figure 2, provide brief explanation of the x and y-axes, and add more context as to what the plot implies.
Author Response
Mahmoud Mohammadi, MD
Department of Medical Oncology
Leiden University Medical Center
Albinusdreef 2
2333 ZA Leiden
The Netherlands
E-mail: m.mohammadi@lumc.nl
January 26, 2025
Dear prof. Dr. Samuel C. Mok and reviewers,
We would like to express our grateful thanks for your valuable comments and suggestions regarding the following manuscript.
Title: Impact of mutation profile on outcomes of neoadjuvant therapy in GIST.
The manuscript has been greatly improved according to your suggestions. We were able to address all the comments of the reviewers (as stated below) and we hereby submit a new revised version of the manuscript with all changes highlighted in yellow.
Looking forward to see your response,
Best regards,
Mahmoud Mohammadi
on behalf of coauthors
Major Comments
- The introduction does not provide the rationale or the importance of studying mutational profiles to overcome the challenges in GIST treatment. Add an adequate explanation of this aspect and connect it to the background. Also, while the paper provides descriptive outcomes based on mutational profiles in GIST patients treated with neoadjuvant imatinib, it does not articulate a clear hypothesis. The objective is presented as "evaluating response and surgical outcomes," but the lack of a central hypothesis weakens the scientific narrative.
- We have revised the introduction to explicitly highlight the rationale and importance of studying mutational profiles in addressing challenges in GIST treatment. Furthermore, we have included a hypothesis.
- What do you mean by "less invasive or lower-risk surgical approaches" (what are your criteria for classification? Elaborate more regarding these concepts.
- These approaches minimize physical trauma, preserve organ function, and reduce perioperative risks compared to traditional open surgery. The classification of these techniques is based on their ability to achieve complete tumor resection while enhancing postoperative recovery and minimizing complications. We have now added this point in the revised version of manuscript (section methods).
- In lines 186-187, when presenting a comparison between patients with KIT-exon 11 mutations and with non-KIT exon 11 mutations in terms of disease progression, the authors did not address the huge sample size difference between the two categories, this skews the interpretation of the results.
- We acknowledge that the sample size difference between the KIT-exon 11 and non-KIT-exon 11 groups may influence the interpretation of results. In the revised manuscript, we have mentioned this limitation.
- The primary outcome (response rate) and secondary outcomes (R0 margin and therapy duration) are not well-defined. For example, how was a "partial response" determined using RECIST 1.1? Was it independently reviewed or assessed by a single investigator?
- The response (stable disease, partial response, complete response, progression disease) is determined by using RECIST 1.1. by a local investigator. Furthermore the primary outcome and secondary outcomes are described in section methods.
- In lines 201-202, the authors write “Negative resection margins (R0) were achieved in 236 patients, which corresponds to 89.8% of those who underwent resection (Table 3)”. Yet they do not break this percentage between KIT and non-KIT mutation groups.
- In table 2, resection margins are shown for KIT-exon 11 group and non-KIT exon 11 group separately. In addition in lines 204-206 the comparison of resection margin between KIT-exon 11 versus non-KIT exon 11 is provided.
- Discussing why the response rates differed significantly across mutational profiles could enrich the “Primary outcome” section.
- We agree that discussing this aspect is important and interesting. We have added a dedicated paragraph in the Discussion section to elaborate on the differences in response rates across mutational profiles, incorporating both molecular and biological perspectives of GIST
- There is somewhat overemphasis on discrepancies between preoperative imaging and resection specimens without signifying its clinical implications.
- We agree that the observed discrepancies do not have direct clinical consequences for current practice. However, it is noteworthy that when assessing the response to neoadjuvant imatinib, the tumor size measured on preoperative imaging often does not correspond to the size of the resected specimen in a significant proportion of patients.
- For KIT-exon 11 mutations, data is given in total but do not differentiate tumor location and size. Do gastric KIT-exon 11 GISTs respond better compared to rectal or small bowel?
- Our study primarily aimed to focus on the overall response of KIT-exon 11 versus non-KIT exon 11 mutations, rather than stratifying by specific tumor locations. As described in the Results section, we observed that the distribution of gastric versus non-gastric GISTs was not significantly different between the two groups, allowing for a more direct comparison of mutational profiles. However, we also noted that non-gastric GISTs were a significant risk factor for non-responders, as highlighted in the Results.
- The paper identifies differences in treatment responses based on mutational profiles but does not sufficiently discuss the underlying molecular mechanisms that might explain these variations:
- Why do KIT exon 11 mutations exhibit better responses and longer durations of neoadjuvant therapy?
- We have expanded the discussion to provide a more in-depth explanation of the molecular mechanisms of better responses of KIT exon 11 mutations.
- How do non-KIT exon 11 mutations contribute to resistance to imatinib?
- Also molecular aspects of less effectiveness of imatinib in non-KIT exon 11 is added in the discussion.
- The potential role of secondary mutations or resistance pathways is not explored.
- We have added the role of secondary mutations and resistance pathways in discussion, as potential resistance mechanism.
- The paper does not address potential tumor heterogeneity within the same patient, which could influence treatment response and surgical outcomes. For example, the presence of mixed mutational subclones or resistance mutations could explain incomplete responses or progressive disease in some cases.
- We have now addressed the role of tumor heterogeneity in influencing treatment response, emphasizing how the presence of mixed mutational subclones or resistance mutations may contribute to incomplete responses or disease progression.
- The paper reports differences in initial imatinib doses for certain mutations (e.g., KIT exon 9 patients started on 800 mg), but it does not analyze how dose escalation or reduction might have influenced response rates, duration of therapy, or surgical outcomes.
- We agree on this insightful comment. Of the 18 patients with KIT exon 9 mutations, 13 received an initial dose of 800 mg, while 5 started on 400 mg. Unfortunately, we were unable to determine whether the dose was subsequently escalated to 800 mg in these 5 patients. Given the small number of patients in this subgroup, we believe their impact on the overall results to be minimal. This limitation has now been acknowledged in the revised manuscript.
- The authors assess the concordance between tumor size reported on preoperative CT scans and pathological resection specimens, and categorize it as a secondary outcome. However, the connection between this analysis and the impact of mutational profiles is not explained anywhere in the manuscript. How is this a relevant outcome to the mutational profile?
- We understand that categorizing the concordance between tumor size reported on preoperative CT scans and pathological resection specimens as a secondary outcome may cause some confusion. We investigated this outcome because, in clinical practice, tumor size assessment on CT scans plays a critical role in evaluating the response to neoadjuvant imatinib and in preoperative planning. Our data demonstrate that in a significant subset of patients, there is a notable discrepancy between tumor size measured on preoperative imaging and pathological resection specimens.
- The study provides valuable insights into mutational profiles and treatment responses but does not sufficiently translate these findings into actionable clinical recommendations. For example:
- Should mutation testing guide decisions on when to transition from neoadjuvant therapy to surgery?
- How should clinicians manage patients with limited or no response to imatinib?
-We appreciate this insightful comment. In response to the suggestions, we have revised the discussion to include actionable clinical recommendations based on our findings. Specifically, we emphasize that mutation testing should guide decisions on when to transition from neoadjuvant therapy to surgery, particularly for patients with non-KIT exon 11 mutations and wild-type KIT/PDGFRA GISTs, who show less favorable responses to imatinib. We also recommend early response assessment using PET-CT imaging at 6-8 weeks to inform the decision of switching to surgical resection if there is limited or no response to neoadjuvant therapy.
Minor Comments
o The abstract does not explicitly mention the study design (retrospective cohort). Adding this information would provide readers with context upfront.
- We added now the study design in the abstract.
o “Gastric GISTs are generally associated with a more favorable prognosis compared to non-gastric GISTs.” is not directly supported by any specific citation in the discussion section.
- Two references are added supporting the favorable prognosis of gastric GIST compared to non-gastric GIST.
o In figure 2, provide brief explanation of the x and y-axes, and add more context as to what the plot implies.
-An explanation is added to figure 2 to clarify its implication. A Bland-Altman plot is provided to illustrate the agreement between tumor size assessed by preoperative CT scans and tumor size measured on resection specimens. In the plot, the x-axis represents the average of the two measurements (CT scan and resection specimen size), while the y-axis shows the difference between the two measurements for each patient. This Bland-Altman plot highlights the extent of variability between the two methods, and it is evident that there is a considerable range of differences in tumor size assessment across patients. The discrepancies between the tumor size measurements were observed in 28.2% of patients.
Reviewer 2 Report
Comments and Suggestions for Authors
The manuscript investigates how different mutational profiles affect the response to neoadjuvant imatinib in patients with GIST. Analyzing data from 326 patients in the Dutch GIST Registry, the study highlights that patients with KIT exon 11 mutations exhibit better partial response rates and higher chances of achieving negative surgical margins (R0) compared to those with non-KIT exon 11 mutations. The findings underscore the variability in treatment response, reinforcing the importance of mutation-specific therapeutic strategies.
The manuscript’s strengths lie in its large patient cohort, real-world clinical relevance, and focus on personalized medicine. However, the retrospective nature of the study introduces potential biases, and the lack of long-term survival data limits the scope of the conclusions. Overall, this study may have some relevance to oncologists and surgeons managing GIST, as it provides valuable insights into optimizing neoadjuvant treatment strategies based on genetic profiling.
Some suggestions:
- consider expanding on the clinical implications of mutation-specific responses, particularly in guiding surgical decisions for borderline resectable GISTs.
- Clarify limitations by detailing potential biases from the retrospective design and lack of survival data.
- Including more discussion on surgical outcomes
Author Response
Mahmoud Mohammadi, MD
Department of Medical Oncology
Leiden University Medical Center
Albinusdreef 2
2333 ZA Leiden
The Netherlands
E-mail: m.mohammadi@lumc.nl
January 26, 2025
Dear prof. Dr. Samuel C. Mok and reviewers,
We would like to express our grateful thanks for your valuable comments and suggestions regarding the following manuscript.
Title: Impact of mutation profile on outcomes of neoadjuvant therapy in GIST.
The manuscript has been greatly improved according to your suggestions. We were able to address all the comments of the reviewers (as stated below) and we hereby submit a new revised version of the manuscript with all changes highlighted in yellow.
Looking forward to see your response,
Best regards,
Mahmoud Mohammadi
on behalf of coauthors
The manuscript investigates how different mutational profiles affect the response to neoadjuvant imatinib in patients with GIST. Analyzing data from 326 patients in the Dutch GIST Registry, the study highlights that patients with KIT exon 11 mutations exhibit better partial response rates and higher chances of achieving negative surgical margins (R0) compared to those with non-KIT exon 11 mutations. The findings underscore the variability in treatment response, reinforcing the importance of mutation-specific therapeutic strategies.
The manuscript’s strengths lie in its large patient cohort, real-world clinical relevance, and focus on personalized medicine. However, the retrospective nature of the study introduces potential biases, and the lack of long-term survival data limits the scope of the conclusions. Overall, this study may have some relevance to oncologists and surgeons managing GIST, as it provides valuable insights into optimizing neoadjuvant treatment strategies based on genetic profiling.
Some suggestions:
- consider expanding on the clinical implications of mutation-specific responses, particularly in guiding surgical decisions for borderline resectable GISTs.
- Thank you for this insightful suggestion. We have expanded the discussion to include the clinical implications of mutation-specific responses in guiding surgical decisions for borderline resectable GISTs.
- Clarify limitations by detailing potential biases from the retrospective design and lack of survival data.
- We appreciate the reviewer's comment regarding the need for further clarification of the study's limitations. In the manuscript, we have acknowledged the retrospective nature of our study, which may introduce biases and confounding factors. Additionally, we addressed the differences in sample sizes between the KIT-exon 11 and non-KIT-exon 11 groups and highlighted the lack of survival and recurrence data.
- Including more discussion on surgical outcomes
-We have now expanded the discussion section by exploring surgical outcomes and related considerations
Round 2
Reviewer 1 Report
Comments and Suggestions for Authors
Thank you for taking all comments into consideration, the manuscript is better now.
I have one comment left regarding point 12: Thank you for providing the clinical significance but the response is not relevant to the comment. The discrepancy between tumor size in imaging and pathological specimens and its implication in evaluating response to imatinib is understood. However, highlighting this discrepancy as a "secondary outcome" is still poorly understood. Explain how does mutational profile affect or relate to this discrepancy.
Thank you,
Author Response
I have one comment left regarding point 12: Thank you for providing the clinical significance but the response is not relevant to the comment. The discrepancy between tumor size in imaging and pathological specimens and its implication in evaluating response to imatinib is understood. However, highlighting this discrepancy as a "secondary outcome" is still poorly understood. Explain how does mutational profile affect or relate to this discrepancy.
We appreciate your insightful comment and the opportunity to clarify our rationale further.
Upon analysis, we did not find a significant association between mutational profile and the frequency of discrepancy between tumor size on preoperative imaging and pathological specimens. We acknowledge that presenting this as a secondary outcome may have led to confusion. Therefore, we have decided not to define it as an outcome measure in the manuscript.
However, given its clinical relevance—particularly in the context of surgical planning and response assessment—we have retained the findings related to tumor size discrepancy in the Results section. We believe this information remains valuable for clinicians managing patients with GIST.
These changes have been implemented in the revised version of the manuscript.